# Unilateral Pleural Effusion after Third Dose of BNT162b2 mRNA Vaccination: Case Report

**DOI:** 10.3390/jpm13030391

**Published:** 2023-02-23

**Authors:** Nikolaos K. Athanasiou, Archontoula Antonoglou, Marios Ioannou, Edison Jahaj, Paraskevi Katsaounou

**Affiliations:** Pulmonary and Respiratory Failure Department, First ICU, Evangelismos Hospital, 106 76 Athens, Greece

**Keywords:** Pfizer vaccine, COVID-19 vaccine, BNT162b2, mRNA vaccination, pleural effusion

## Abstract

Vaccination remains the best strategy against coronavirus disease 2019 (COVID-19) in terms of prevention. The efficacy and safety of COVID-19 vaccines is supported by well-designed clinical trials that recruited many participants. It is well-known that vaccination is associated with local side effects related to the injection site, and mild, systemic side effects. However, there has been an increase in the occurrence of what is known as infrequent adverse effects in the population of vaccinated individuals in real life. We present the case of a 46-year-old woman with no past medical history, who presented with a sharp chest pain with deep inspiration, a few days after receiving the third dose of the Pfizer-BioNTech COVID-19 mRNA vaccine (BNT162b2). There is an association between the BNT16b2 vaccination and myocarditis, pericarditis, and even bilateral pleural effusions. To the best of our knowledge, this is the first report featuring a unilateral pleural effusion in a patient with no known past medical history, who did not develop cardiac involvement nor have any viral infection. The aim of our report is to inform health professionals of the possibility of encountering this rare adverse event in their daily practice, as the population of individuals who are receiving additional vaccine doses is increasing steadily.

## 1. Introduction

The clinical spectrum of coronavirus disease 2019 (COVID-19) is heterogeneous, ranging from a flu-like syndrome to severe pneumonia that may progress to acute respiratory distress syndrome (ARDS). It has been responsible for more than 6.5 million deaths worldwide and vaccination remains the best form of prevention currently available [1,2]. The new messenger ribonucleic acid (mRNA) vaccines consist of a segment of viral mRNA enveloped by lipid nanoparticles to activate the immune system. As of now, many mild systemic reactions (such as fatigue, headache, muscle pain, fever) and local reactions (such as injection site pain, injection site swelling) have been reported, as well as cases of rare/unusual adverse events, in the context of the ongoing global campaign [2,3,4]. Rare adverse events, such as myocarditis and pericarditis, in addition to bilateral pleural effusions, have been described in the literature as being associated with vaccination with the Pfizer-BioNTech COVID-19 mRNA vaccine (BNT162b2) [4,5,6]. Whilst new pandemic outbreaks emerge periodically, and vaccinations worldwide are ongoing, further booster doses are given to the general population, and subsequently, new adverse effects are described. Here we present the case of a 46-year-old woman, who arrived at the emergency department with acute pleuritic chest pain ten days after receiving the third dose of BNT162b2, with the objective of proving a causal relationship between the vaccination against COVID-19 and the development of a unilateral pleural effusion. All of this, in a patient with no known past medical history, no concurrent cardiac involvement or any infection. To the best of our knowledge, there is no similar case report that has been described in the literature at this point in time.

## 2. Case Report

In early November 2021, a 46-year-old woman presented to the emergency department of our hospital with an episode of acute onset, sharp shingles chest pain, particularly during deep inspiration and the associated shortness of breath. She did not report a cough, shortness of breath or any prodrome symptoms, similar to acute upper or lower respiratory tract infection. From the medical history that we were given, we ascertained that she was a mother of two children (uneventful pregnancies and caesarean deliveries), she has never smoked, she denied any occupational exposures and had no previous medical history. The patient did not report any arthralgias or any family history of myocardial infarction or thrombophilia. In January 2021, she had two inoculations of the BNT162b2 mRNA vaccine, with an interval between the doses of 21 days.

Ten days prior to symptom onset, she had received a third dose of the BNT162b2 mRNA vaccine. She did not report any adverse effects after vaccination except intense discomfort at the injection site. She stated that she had not been infected with the SARS-CoV-2 virus since the start of the pandemic and due to her profession, she had regular checkups with real-time PCR of nasopharyngeal swabs that were consistently negative. Moreover, the patient reported that four days before the third dose of the COVID-19 vaccine, she was tested for immunoglobulin G (IgG) against SARS-CoV-2 nucleocapsid (NC) proteins, ruling out any physical illness. Furthermore, she was tested for anti-spike immunoglobulin G (IgG) antibody titers, whereby levels are expressed in arbitrary units (AU)/mL, which were 661 AU/mL. On the same day, she underwent laboratory testing with full blood count, basic metabolic panel, liver panel, coagulation panel including D-dimer testing, and thyroid hormone panel, all of which were within the normal range. The patient brought all the aforementioned test results on her admission. According to the external laboratory in which the test was conducted, levels of circulating SARS-CoV-2 anti-spike IgG (S) and anti-nucleocapsid IgG (NC) antibodies were quantified using the Abbott Diagnostics SARS-CoV-2 IgG chemiluminescent microparticle immunoassay (Abbott Diagnostics, Abbott Park, Illinois) on an Abbott Diagnostics Architect i2000 SR, according to the manufacturer’s instructions. For anti-spike (S), results were expressed in AU/mL and were interpreted as positive if ≥50, and for anti-nucleocapsid IgG (NC) antibodies the index values of 1.4 S/C and above are considered positive per the manufacturer’s instructions [7].

On admission, she was afebrile (36.2 °C), her blood pressure was 115/65 mm Hg, her pulse was regular at 110 beats per minute, her oxygen saturation was 97% in ambient air, and she had a respiratory rate of 22 breaths per minute. Physical examination of the chest revealed a small reduction of vesicular sounds on the left pulmonary base, there were no audible cardiac murmurs and the abdominal examination was unremarkable. There were no other findings from the rest of the clinical examination, including no evidence of palpable lymphadenopathy.

The laboratory test results were as follows: white blood cell count (11.110 cells/mm^3^, normal range: 4–10,500 cells/mm^3^) (neutrophils, 45.6%; lymphocytes, 45.9%; eosinophils, 1.6%); hemoglobin (12.6 g/dL, normal range: 12–15 g/dL); platelet count (308.000 /mcL, normal range: 140–450.000 /mcL); C-reactive protein (0.5 mg/dL, normal range: <0.5 mg/dL); lactate dehydrogenase (282 IU/L, normal range: <225 IU/L); creatine kinase (91 IU/L, normal range: 10–173 IU/L); creatine kinase-myocardial band (30 IU/L, normal range: 1–18 IU/L); international normalized ratio 1.06; activated partial thromboplastin time (30.8 s, normal range: 26–38 s); fibrinogen (300 mg/dL, normal range: 200–400 mg/dL); D-dimer (0.66 μg/mL, normal range: <0.5 μg/mL); troponin T highly sensitive (2 pg/mL, normal range: <12 pg/mL); NT-proB-type natriuretic peptide (53 pg/mL, normal range: <450 pg/mL). Renal, liver, and pancreatic function tests were within the range of normality. A nasopharyngeal swab was taken that tested negative for SARS-CoV-2 via real-time PCR. The real-time PCR test was performed on the VIASURE, Real Time PCR Detection Kits (by CerTest BIOTEC) according to the manufacture’s instruction, who report high sensitivity and specificity by this particular method.

During her stay in the emergency department the 12-lead electrocardiogram showed normal sinus rhythm and the cardiac ultrasound was without any pathological findings. A focused lung ultrasound was conducted without the presence of B-lines, but a small anechoic left pleural effusion was discovered when placing the transducer in the left posterior axillary line at the lower part of the chest (Figure 1). Due to the left sharp “shingles-like” pain and the pleural effusion that was found, an abdomen ultrasound was performed in order to investigate possible abdominal pathology, which was unremarkable. According to the Wells Criteria for Pulmonary Embolism, she was a moderate risk patient for pulmonary embolism. Due to the absence of any other obvious cause of the pleural pain and the pleural effusion, a computed tomography pulmonary angiography (CTPa) was conducted. The CTPa scan was negative for pulmonary embolism, but verified the aforementioned left pleural effusion. The CTPa did not show any other significant findings from the pulmonary parenchyma or the mediastinum, besides a large diaphragmatic hernia already known about (Figure 2).

The patient was admitted for observation for a 24 h period and in order to run further tests. While working to establish a differential diagnosis, a nasopharyngeal swab respiratory panel was sent that included testing for viruses (influenza A, A/H1, A/H3, B; Adenovirus; coronavirus HKU1, NL63, 229E, OC43; parainfluenza virus types 1, 2, 3, 4; RSV; human rhinovirus/enterovirus; human metapneumovirus, Middle East respiratory syncytial coronavirus; SARS-CoV-2) and bacteria (bordetella pertussis, bordetella parapertussis, Chlamydia pneumoniae, Mycoplasma pneumoniae). Furthermore, screening and laboratory testing for autoimmune diseases with antinuclear antibodies (ANA), anti-double-stranded DNA (anti-dsDNA), rheumatoid factor (RF), immunoglobulin G4 (IgG4), anti-cyclic citrullinated peptide (anti-CCP), and QuantiFERON and electrophoresis of proteins, were within the normal values. The varicella-zoster virus antibody immunoglobulin G (IgG) that were sent yielded positive results, whilst IgM antibodies were negative. We tested the anti-spike IgG antibody titers after the third vaccination, which were 25.600 AU/mL, were determined by chemiluminescent microparticle immunoassay (Abbott Diagnostics, Abbott Park, Illinois) on an Abbott Diagnostics Architect i2000 SR, according to the manufacturer’s instructions. Antibodies levels were expressed in arbitrary units (AU)/mL and results ≥50 AU/mL were considered as positive [7].

Due to the pleural effusion’s small volume and the patient’s refusal to take part in any invasive procedure, diagnostic thoracic paracentesis was not carried out. Thus, we could not collect valuable information from pleural fluid analysis.

The patient received symptomatic therapy with painkillers and only five days later the pain improved. Throughout her admission, there were no signs or symptoms consistent with a focus of infection, whether respiratory or otherwise. On the seventh day follow-up, we repeated the focused lung ultrasound and the pleural effusion was not located, indicating a full remission. The follow-up cardiac ultrasound was unremarkable. A nasopharyngeal swab was taken and sent for a full respiratory panel, which did not yield any positive results. All follow-up blood work returned within the normal ranges. Twenty days after her discharge she returned in order to collect the remaining laboratory test results, and our patient was tested for IgG antibodies against the SARS-CoV-2 nucleocapsid protein, which was negative. Immunoglobulin class G antibodies to the nucleocapsid protein of SARS-CoV-2 was performed on an Alinity i analyzer (Abbott Diagnostics, Abbott Park, IL, USA), which is a chemiluminescent microparticle immunoassay, and the index values of 1.4 S/C and above are considered positive per the manufacturer’s instructions. Using an index S/C threshold of 1.4, the manufacturer reported a sensitivity of 86.4% after 7 days from symptom onset and 100% after 14 days, and a specificity of 99.6%, using RT-PCR as the gold standard [7].

The final patient follow-up was at the 1-year mark; up to October 2022 she had continued without any observation of a relapse of the pleural effusion or any other symptom that indicates systematic disease. She also presented a normal mammography, and a thyroid ultrasound without any pathological findings. Consequently, in our opinion, the cause of the pleural effusion was related to the COVID-19 vaccination.

## 3. Discussion

A pleural effusion is an excessive accumulation of fluid in the pleural space. Pathophysiologically, it occurs due to increased production and increased permeability from the capillaries of the pleura and the pulmonary interstitium and, at the same time, it is due to decreased lymphatic drainage. It is estimated that pleural effusion develops in more than 1.5 million patients each year in the United States, with a wide range of causes of unilateral pleural effusion in adults, usually as a consequence of a wide variety of disorders of the lungs, pleura and systemic disorders [8,9]. Patients most commonly present with shortness of breath, initially on exertion, a predominantly dry cough, and pleuritic chest pain. Pleural pain is often described as sharp, stabbing and made worse by deep respiration. The localization of pain is indicative of the site of the underlying pathological process, while the time of symptom onset is essential to the differential diagnostic approach. Pleuritic pain with sudden onset suggests diagnosis of pneumothorax, pulmonary emboli or pulmonary infraction, whereas pleuritic pain building over a few hours may suggest infection, such as pneumonia or pleurisy. Onset over several days suggests empyema, malignancy or tuberculosis. The large majority of cases of pleural effusion arise from congestive heart failure, pneumonia, and cancer. Thoracentesis is such a frequently performed procedure that it may be diagnostic and/or therapeutic. Diagnostic thoracentesis should be performed on almost all patients with a pleural effusion of unknown origin. Its main purpose is to differentiate between transudate and exudate fluid. As a result, the patient’s history, thoracocentesis results, laboratory testing and diagnostic imaging (e.g., computed tomography, ultrasound), assist in the identification of the underlying cause and the approach to its management [8,9]. Idiopathic pleural effusions are reported to account for 5–25% of cases in different studies after thorough investigation and the performing of biopsies when indicated [10].

As the number of vaccinated individuals increases, and booster doses are given through ongoing vaccination campaigns, so does the occurrence of adverse events observed in the community. Our patient developed unilateral pleural effusion after receiving the third dose of the COVID-19 mRNA vaccine. Pleural effusions have been noted in only a minority of severe COVID-19 cases (up to 5%) [11]. The pathophysiology behind the formation of a pleural effusion is unclear, but the binding of the SARS-CoV-2 virus through the angiotensin-converting enzyme 2 (ACE2) receptor present in lung tissue leading to direct tissue damage, may be part of the underlying mechanism. The presence of a marked decrease in lymphocytes, increased platelets, CRP, LDH, and D-dimers levels in COVID-19 patients with pleural effusion, suggest the role of systemic inflammatory response syndrome and the cytokine storm as the pathological mechanism behind pleural effusion in COVID-19 [11,12,13]. Despite all the published literature accumulated so far, there is no mention of isolated pleural effusion in healthy individuals receiving the BNT162b2 vaccine without the concurrent findings of myocarditis and/or pericarditis.

Myocarditis and pericarditis with or without coexisting pleural effusions are some of the rarer adverse events encountered in individuals receiving the BNT162b2 vaccination, and if encountered, are usually observed in men under 40 years old in the first week following injection of the second primary series dose or first booster dose, with most booster doses likely administered at least 5 months after completing the primary vaccination [4,5,6,14,15,16,17]. Additionally, a rare case of latent herpes zoster reactivation has been published, linked to the administration of the mRNA COVID-19 vaccine, where in rare cases the presence of VZV can be associated with pleural effusions [18,19]. Additionally, a case involving IgG4 related pleural disease developed after the COVID-19 mRNA vaccination has been described recently, but it is known whether the disease itself may be a cause of the pleural effusions [20]. Finally, a case has also been observed involving bilateral pleural effusions and polyarthritis in an elderly patient with comorbidities developed after receiving the BNT162b2 vaccine [21]. The main causes of acute chest pain and shortness of breath are listed in Table 1, as they could help health professionals in diagnosing causes of acute chest pain and shortness of breath after vaccination with BNT162b2 [22].

The mechanism underlying the development of pleural effusions after receiving an mRNA COVID-19 vaccine is currently unknown. We made the diagnosis of vaccine-induced pleuritis by exclusion. Firstly, the patient did not display signs or complain of symptoms indicating an upper or lower respiratory tract infection, the thoracic CT did not show any lesions or involvement of the lung parenchyma, while two nasopharyngeal respiratory panels (including testing for SARS-CoV-2) seven days apart were both negative, and IgG antibodies against the SARS-CoV-2 nucleocapsid protein were negative 20 days after discharge. It has been reported that negative PCR tests have been documented as false negatives SARS-CoV-2 infected patients. In many cases, this is a consequence of incorrect technique in the collection of the sample. In our case, every care has been taken in our practice to utilize the correct and indicated technique in the collection of these samples (according to international guidelines) and the subsequent analysis of this material by the laboratory has a documented highly sensitive and specific profile that is mentioned above [23]. On the other hand, we documented the NC antibody titers 20 days after the development of the pleural effusion, in the exceedingly rare event that it was due to physical illness with two false negative PCR tests. According to current data the detection of NC antibodies 14 days after infection is a highly specific finding [7]. It should be noted that the patient had no typical presentation nor a family history of autoimmune diseases, and the tests for autoantibodies were within the normal ranges. In addition to pleural effusion, there was no laboratory or cardiac ultrasound evidence of myocarditis or pericarditis. The thyroid function tests were normal, a pulmonary embolism was excluded, and on an outpatient basis a mammography and thyroid ultrasound were conducted, in addition to the follow-up at one year, all with unremarkable findings.

## 4. Conclusions

Thus, we come to the conclusion that the BNT162b2 mRNA vaccination may be associated with unilateral pleural effusion in healthy individuals without the concurrent diagnosis of myocarditis or pericarditis, nor any infection. Only paracetamol as pain relief was provided to our patient and the pleural effusion resolved within a week. To our knowledge this is the first case of unilateral pleural effusion described after vaccination with BNT162b2. We believe that this case report is worth publishing, as it provides a broadened differential diagnostic list to the causes of acute chest pain and shortness of breath post-COVID-19 vaccination with BNT162b2. As new mutations of COVID-19 arise and millions of people globally are vaccinated with booster doses, it is of great importance that conditions induced by the vaccination are diagnosed and reported. However, this does not underestimate the value of vaccination as an effective, and at this point, essential strategy against COVID-19.

## Figures and Tables

**Figure 1 jpm-13-00391-f001:**
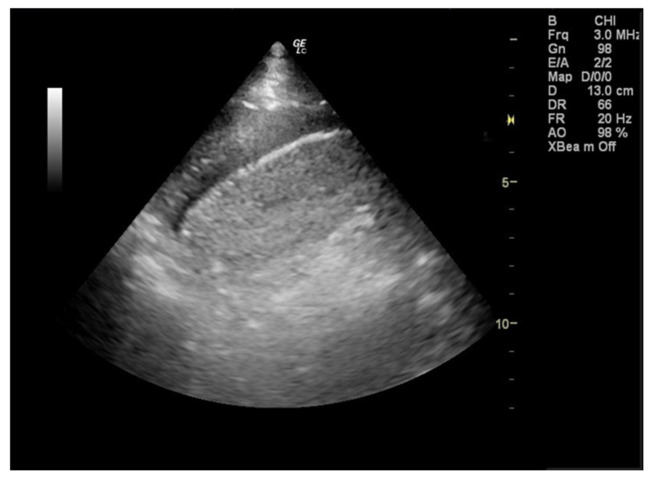
During the focused lung ultrasound a small left pleural effusion was detected by placing the transducer in the left posterior axillary line at the lower part of the chest (zone L3).

**Figure 2 jpm-13-00391-f002:**
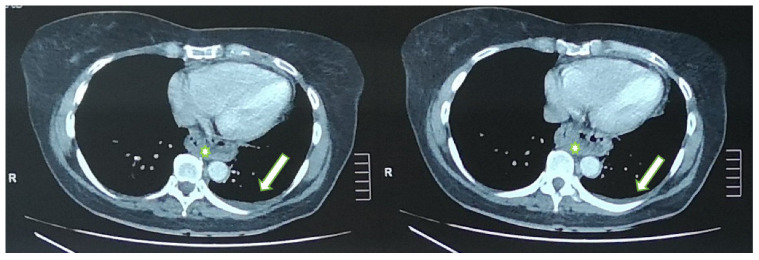
Axial CT images show the small left pleural effusion (arrow); diaphragmatic hernia (asterisk).

**Table 1 jpm-13-00391-t001:** Main causes of acute chest pain and shortness of breath.

Pneumonia and/or pleural infection (usually with a fever and cough) Pulmonary embolism Pneumothorax Acute pleural effusion (i.e., due to systemic erythematous lupus) Myocardial infarction Aortic dissection Arrhythmias (causing angina) Esophageal perforation/rupture

## Data Availability

The data presented in this study are available on request from the corresponding author.

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
