# Peer review of "Unilateral Pleural Effusion after Third Dose of BNT162b2 mRNA Vaccination: Case Report"

_jpm, 2023, doi:10.3390/jpm13030391_

Round 1
Reviewer 1 Report
Authors reported a case of unilateral pleural effusion after third dose of BNT162b2 mRNA 2 vaccination.
Patient was tested for anti-spike IgG antibodies against SARS-CoV-2 after the third dose of vaccine. What about testing for IgG against SARS-CoV-2 nucleocapsid protein?
Did authors test this patient during or after hospitalization for the presence of IgG against SARS-CoV-2 nucleocapsid protein after the third dose of vaccine?
Author Response
Thank you for bringing this into our attention. Indeed, the patient twenty days after her discharge collected the answers of the remaining laboratory results and was tested for anti-spike IgG antibodies against SARS-CoV-2 nucleocapsid protein which was negative.
Reviewer 2 Report
In this study, Athanasiou et al. reported unilateral pleural effusion after the third dose of the BNT162b2 mRNA COVID-19 vaccine. This report could have significant importance to clinicians closely monitoring the field. It could also be a guide for patients and health professionals for possible causes of acute chest pain and shortness of breath post-COVID-19 vaccination with BNT162b2. I, therefore, advise for its acceptance after a minor language editing.
Author Response
Thank you for your positive comments. The manuscript was edited from a native English speaker. We additionally added a table with the main causes of acute chest pain and shortness of breath in order to help health professionals in differential diagnosing acute chest pain and shortness of breath post-COVID -19 vaccination with BNT162b2.